# No-Tillage Combined with Appropriate Amount of Straw Returning Increased Soil Biochemical Properties

**Wanhua Chen [1,2], Wei Yuan [1,2], Jie Wang [1,2], Ziyang Wang [1,2], Zhengping Zhou [1,2] and Shiping Liu [1,2,*]**

[1] Jiangsu Key Laboratory of Crop Genetics and Physiology/Jiangsu Key Laboratory of Crop Cultivation and Physiology, Agricultural College of Yangzhou University, Yangzhou 225009, China; cwh970521@163.com (W.C.); yw1352917010@126.com (W.Y.); wjdizi1111@163.com (J.W.); wzy1155665@162.com (Z.W.); zpingzhou@foxmail.com (Z.Z.)

[2] Jiangsu Co-Innovation Center for Modern Production Technology of Grain Crops, Yangzhou University, Yangzhou 225009, China

* Correspondence: spliu@yzu.edu.cn; Tel.: +86-0514-87979351

**Abstract:** (1) Background: Few studies have focused on the interaction of tillage and straw returning on soil carbon and nitrogen. Therefore, this study was conducted for investigating the effects of tillage and straw returning on soil biochemical properties under a rice–wheat double cropping system; (2) Methods: Six treatments were set up to determine soil biochemical properties, including no-tillage with all straw returning (NTS), wheat plow tillage and rice no-tillage with half straw returning (RT1), wheat no-tillage and rice plow tillage with half straw returning (RT2), plow tillage with all straw returning (CTS), less tillage with half straw returning (MTS), and plow tillage with no straw returning (CT); (3) Results: Straw returning increased soil microbial biomass carbon (SMBC) and soil microbial biomass nitrogen (SMBN), but had no significant effects on total nitrogen (TN) and soil organic carbon (SOC). In the treatments of straw returning, the contents of SMBC, SMBN, TN, and SOC under no-tillage were increased in the 0–7 cm soil layer. Tillage and straw returning had no significant effects in the 7–14 cm and 14–21 cm soil layers. In addition, SMBC/SMBN for all the treatments was maintained within a reasonable range, and microbial quotient (SMQ) and SMBN/TN in the no-tillage treatment had a significant improvement; (4) Conclusions: The results showed that no-tillage with an appropriate amount of straw returning improved the soil biochemical properties and maintained the nitrogen mineralization capacity in the 0–7 cm soil layer for this region.

**Keywords:** tillage; straw returning; soil biochemical properties; soil depth; rice growth stages

## 1. Introduction

Crop production plays a critical role in providing food supply and ensuring food security. With the advancement of crop production technology and agricultural machinery, grain yield has been increasing continuously. However, the problem of straw wastes has also become increasingly prominent with the increase in grain yield [1,2]. In 2016, the total amount of agricultural straws reached 980 million tons in China, but the comprehensive utilization rate of straw was only 81.68% [3]. There are large differences in the level of straw utilization between regions. The phenomena of burning straw in the fields and straw discarding still exist in most parts of the country [4,5]. Recycling and utilization of straw have now become a severe challenge. Crop straw returning to the fields is one of the most efficient measures to solve the problem of straw waste [6]. As an agricultural resource that can replace chemical fertilizers [7,8], crop straws can alleviate the environmental pollution caused by chemical fertilizers to a certain extent. Many scholars found that effective straw returning had optimized the ecological environment of farmlands, improved soil physical and chemical properties, and increased crop yields [9–11]. Soil provides essential nutrient elements for the growth and development of crops [12]. Changes in the soil environment can affect the growth of crop plants. Tillage is an important factor affecting

soil environment and quality [13]. The current common tillage methods are no-tillage, reduced tillage, plow tillage, rotary tillage, deep tillage, etc. Conventional tillage methods may destroy soil aggregate structure and reduce soil quality [14]. In contrast, conservation tillage can increase nutrient content and microbial quantity in the 0–7 cm soil layer, thereby improving the soil environment. Straw returning to the fields combined with appropriate tillage can effectively improve the farmland environment and increase grain yield on the premise of solving the problem of straw waste [15].

Soil organic carbon plays a critical role in regulating the flow of soil nutrients and improving the physical structure of the soils [16]. Soil microbial biomass carbon (SMBC) refers to the organic carbon contained in soil microorganisms, which is responsible for the decomposition of organic matter and is an important indicator of soil fertility [17]. Most of the nitrogen in the soils exists in the form of organic nitrogen, accounting for 92–98% of total nitrogen. Soil microbial biomass nitrogen (SMBN) is the most vulnerable component of soil organic nitrogen to the environment and tillage measures [18]. Crop straw and tillage are directly related to the changes in soil carbon pool and the efficiency of nitrogen conversion in farmland ecosystems [19–21]. Decomposition of straw makes the carbon in the straw return to the soils [22], and different tillage methods make organic matter fixed in the soils, which can effectively reduce the loss of carbon and nitrogen in farmlands [23,24].

At present, most scholars have only conducted short-term studies on farmland soil carbon and nitrogen in China with a single method of tillage or straw returning [25]. There is little knowledge on the interaction between tillage and straw returning on soil carbon and nitrogen. We hypothesized that different tillage methods and straw returning practices may have diverse effects on soil carbon and nitrogen.

This study was a one-year field experiment at a long-term experiment that did not involve the measurements of soil biochemical properties before 2020. So we set up this study to analyze the effects of tillage and straw returning on the characteristics of total nitrogen (TN), soil organic carbon (SOC), soil microbial biomass carbon (SMBC), and soil microbial biomass nitrogen (SMBN). The objective of this study was to find a suitable farming practice for both increasing wheat and rice grain yield and improving the farmland ecological environment.

## 2. Materials and Methods

### 2.1. Experimental Site

This study was conducted during the rice-growing season of 2020 (from May 2020 to October 2020) at a long-term experimental location starting in November 2001 in the experimental field of Yangzhou University in Yangzhou, Jiangsu Province, China ($32°23'36''$ N, $119°24'51''$ E). This location belongs to a northern subtropical humid climate zone. It has an annual average temperature of 14.8–15.3 °C, an annual precipitation of 961–1048 mm, and an annual sunshine duration of 1896~2182 h. Since 2001, a wheat–rice double cropping system has been used at this location. The soil was a sandy loam and tested containing 17.66 g/kg organic matter, 1.07 g/kg total nitrogen, 80.6 mg/kg alkali hydrolyzable nitrogen, 22.6 mg/kg available phosphorus, and 95.5 mg/kg available potassium in November 2001.

### 2.2. Experimental Design

In this study, Nanjing 9108, a locally widely-extended rice variety, was used. There were 6 treatments, including no-tillage with all straw returning (NTS), wheat plow tillage and rice no-tillage with half straw returning (RT1), wheat no-tillage and rice plow tillage with half straw returning (RT2), plow tillage with all straw returning (CTS), less tillage with half straw returning (MTS), and plow tillage with no straw returning (CT) (Table 1). For each treatment, there were 3 replicates. The study was designed as a single-factor experiment. Each plot was 8 m long, 6.25 m wide, and had an area of 50 $m^2$.

For the NTS and RT1 treatments, the rice seeds were directly sown on 3 June 2020. For the RT2, CTS, MTS, and CT treatments, the rice seeds were sown on 22 May 2020 and

cultured at a seedling bed, and transplanted to the paddy field on 14 June 2020. The field practices of all the treatments, including fertilizer application, irrigation, weed, and pest control, were conducted in conformity with local recommendations. We were advised to apply N at the rate of 245 kg ha$^{-1}$ from urea, P at the rate of 600 kg ha$^{-1}$ from calcium superphosphate, and K at the rate of 120 kg ha$^{-1}$ from potassium chloride before sowing. Additionally, 147 kg urea ha$^{-1}$ should be applied in the late growth stage of rice.

**Table 1.** Experimental design: different tillage methods and the amount of straw returning.

| Treatment | Tillage Methods | The Amount of Straw Returning |
|---|---|---|
| NTS | No-tillage with all straw returning | 4500 kg ha$^{-1}$ per season |
| RT1 | Wheat plow tillage and rice no-tillage with half straw returning | 3000 kg ha$^{-1}$ in the rice season |
| RT2 | Wheat no-tillage and rice plow tillage with half straw returning | 3000 kg ha$^{-1}$ in the wheat season |
| CTS | Plow tillage with all straw returning | 4500 kg ha$^{-1}$ per season |
| MTS | Less tillage with half straw returning | 3000 kg ha$^{-1}$ per season |
| CT | Plow tillage with no straw returning | No straw returning |

*2.3. Measurements*

Soil samples in the field were taken at the growth stages of heading, grain filling, and maturity of rice. The three-point sampling method is used to take soil samples from three soil depths: 0–7 cm, 7–14 cm, and 14–21 cm (The cultivated layer was generally 20 cm. The depth of less tillage was 8–10 cm, and that of plow tillage was 14–16 cm. Therefore, the cultivated layer was divided into 0–7 cm, 7–14 cm and 14–21 cm for research).

Soil samples were dried and passed through a 100-mesh sieve for the determination of soil fertility. The total nitrogen content was determined using the Kjeldahl method [26], and organic carbon concentration was determined using the potassium dichromate volumetric method [27].

Microbial biomass carbon and nitrogen were determined using the chloroform fumigation-$K_2SO_4$ extraction method [28]. The ratio of soil microbial biomass carbon to nitrogen is defined as the ratio of SMBC to SMBN, namely SMBC/SMBN. The microbial quotient is the ratio of SMBC to SOC, namely SMQ. The ratio of microbial biomass nitrogen to total nitrogen is SMBN/TN.

*2.4. Statistical Analysis*

The data of each parameter were subjected to analysis of variance (ANOVA) with the statistical package with SPSS 16.0 according to the single-factor randomized design. The differences in SMBC, SMBN, SOC, TN, and the ratio of each biomass between different treatments were compared based on an ANOVA-protected LSD0.05 test.

**3. Results**

*3.1. The Content of SMBC at the Late Growth Stages of Rice*

The change of SMBC content at the late stage of rice growth showed a unimodal trend. The content of SMBC decreased with the increase in soil depth and it was mainly determined in the 0–7 cm layer. SMBC content ranged from 522.6 mg/kg to 1077.5 mg/kg in the 0–7 cm soil layer, from 358.3 mg/kg to 839.9 mg/kg in the 7–14 cm soil layer, and from 252.7 mg/kg to 679.8 mg/kg in the 14–21 cm soil layer (Figure 1). In the 0–7 cm and 7–14 cm soil layers, the NTS treatment had the highest SMBC content and the CT treatment had the lowest. In the 14–21 cm soil layer, SMBC content was significantly higher than that in the CTS, MTS, and RT2 treatments. In other treatments, SMBC content followed a descending order of RT1, NTS, and CT. Continuous no-tillage increased SMBC content in the 0–7 cm and 7–14 cm soil layers to a certain extent, while in the 14–21 cm soil layer, plowing and less tillage were more beneficial to SMBC accumulation. On the whole, straw returning increased SMBC content in all three soil layers (Figure 1).

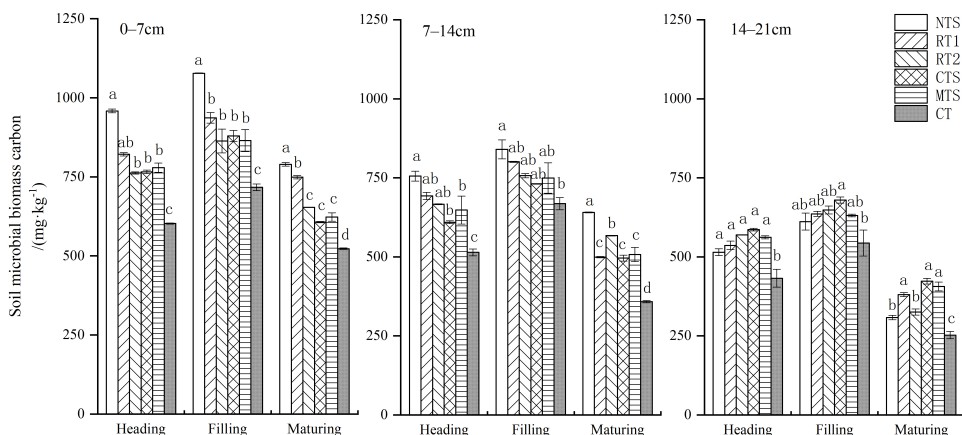

**Figure 1.** The mean and standard error of soil microbial biomass carbon (SMBC) for different treatments during the late growth stages of rice (heading, grain filling, and maturity). The tillage acronyms are NTS = no-tillage with all straw returning, RT1 = wheat plow tillage and rice no-tillage with half straw returning, RT2 = wheat no-tillage and rice plow tillage with half straw returning, CTS = plow tillage with all straw returning, MTS = less tillage with half straw returning, CT = plow tillage with no straw returning. Lowercase letters indicate significant differences among different tillage methods and straw returning amounts.

### 3.2. The Content of SMBN at the Late Growth Stages of Rice

The trend of SMBN content was similar to that of SMBC in terms of stages, treatments, and soil layers. SMBN content ranged from 68.9 mg/kg to 134.1 mg/kg in the 0–7 cm soil layer, from 53.8 mg/kg to 100.8 mg/kg in the 7–14 cm soil layer, and from 36.4 mg/kg to 79.6 mg/kg in the 14–21 cm soil layer (Figure 2). In the 0–7 cm and 7–14 cm soil layers, SMBN content in the NTS treatment was significantly higher than that in other treatments. In the 14–21 cm soil layer, the CTS treatment and the CT treatment had the highest and lowest SMBN content, respectively. As a whole, continuous no-tillage increased SMBN content in the 0–7 cm and 7–14 cm soil layers, while plowing increased SMBN content in the 14–21 cm soil layer. Straw returning increased SMBN content in all the soil layers (Figure 2).

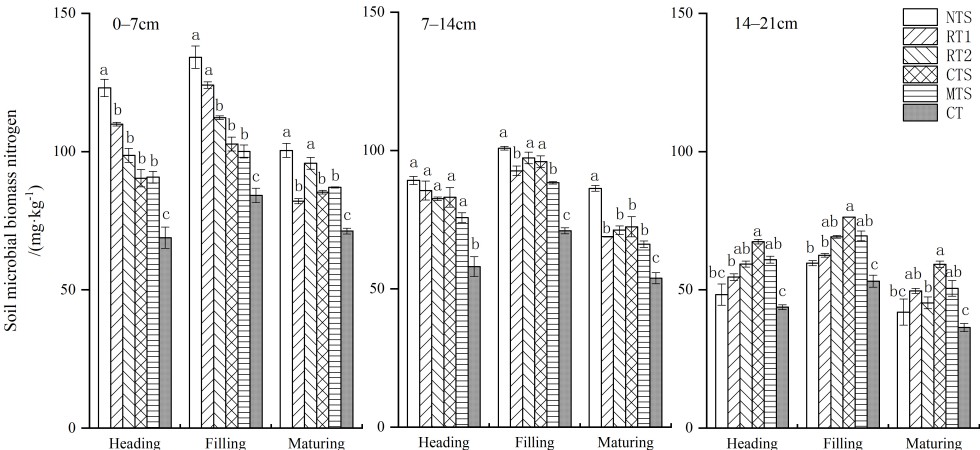

**Figure 2.** The mean and standard error of soil microbial biomass nitrogen (SMBN) for different treatments during the late growth stages of rice (heading, grain filling, and maturity). The tillage acronyms are NTS = no-tillage with all straw returning, RT1 = wheat plow tillage and rice no-tillage with half straw returning, RT2 = wheat no-tillage and rice plow tillage with half straw returning, CTS = plow tillage with all straw returning, MTS = less tillage with half straw returning, CT = plow tillage with no straw returning. Lowercase letters indicate significant differences among different tillage methods and straw returning amounts.

### 3.3. The Content of TN at the Late Growth Stages of Rice

The TN content had no significant changes in terms of soil depth and rice growth stage. In general, TN content varied from 0.7 kg/kg to 1.3 g/kg (Figure 3). In the 0–7 cm soil layer, the NTS and CTS treatments had the highest TN content, while the CT treatment had the lowest TN content. In the 7–14 cm and 14–21 cm soil layers, the TN content of the CTS and MTS treatments was significantly higher at heading and maturity. The TN content of the NTS treatment was lower than that of the CT treatment. At grain filling, the TN content of the NTS treatment was significantly highest, and there was no significant difference in TN content between other treatments. As a whole, continuous no-tillage combined with straw returning effectively increased TN content at grain filling, while plowing and low-tillage increased TN content in the 7–14 cm and 14–21 cm soil layers at heading and maturity. The influence of straw returning on TN content was gradually weakened in the 7–14 cm and 14–21 cm soil layers (Figure 3).

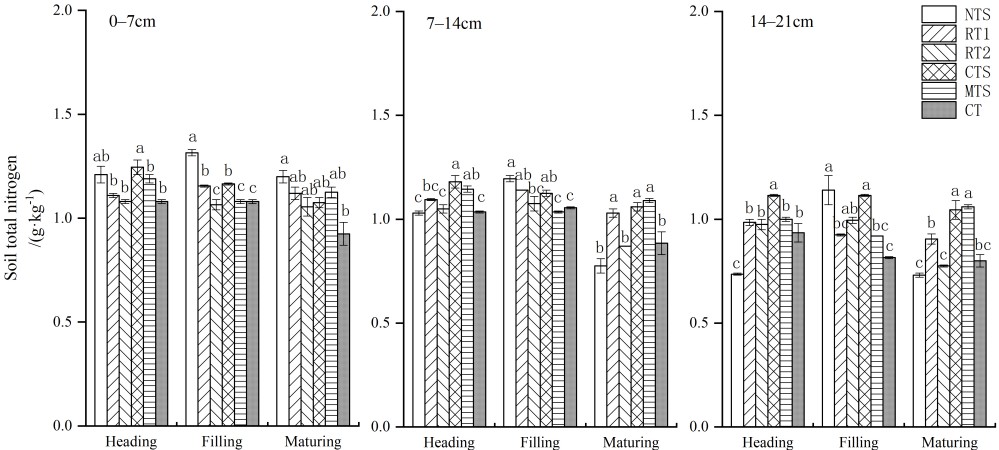

**Figure 3.** The mean and standard error of soil total nitrogen (TN) for different treatments during the late growth stages of rice (heading, grain filling, and maturity). The tillage acronyms are NTS = no-tillage with all straw returning, RT1 = wheat plow tillage and rice no-tillage with half straw returning, RT2 = wheat no-tillage and rice plow tillage with half straw returning, CTS = plow tillage with all straw returning, MTS = less tillage with half straw returning, CT = plow tillage with no straw returning. Lowercase letters indicate significant differences among different tillage methods and straw returning amounts.

### 3.4. The Concentration of SOC at the Late Growth Stages of Rice

There was no significant difference in SOC concentration between heading and grain filling, but SOC concentration rose rapidly at maturity. The difference in SOC concentration between soil depths was not obviously observed. SOC concentration ranged from 12.7 g/kg to 22.0 g/kg in the 0–7 cm soil, from 12.1 g/kg to 18.2 g/kg in the 7–14 cm soil layer, and from 10.8 g/kg to 17.4 g/kg in the 14–21 cm soil layer (Figure 4). Different tillage methods were only observed to have effects on SOC concentration in the 0–7 cm soil layer, especially at grain filling and maturity. At this depth, the NTS treatment had the highest SOC concentration and the CT treatment had the lowest SOC concentration. In the 7–14 cm and 14–21 cm soil layers, there was no significant difference in SOC concentration between heading and grain filling. The SOC concentration of the NTS treatment at maturity was lower than that of the CT treatment. Compared with the 0–7 cm soil layer, SOC concentration in the 7–14 cm and 14–21 cm soil layers decreased by 31.0% and 38.2%, respectively. Long-term no-tillage with straw returning increased SOC concentration in the 0–7 cm soil layer. In the 7–14 cm and 14–21 cm soil layers, straw returning had no significant effect on SOC concentration (Figure 4).

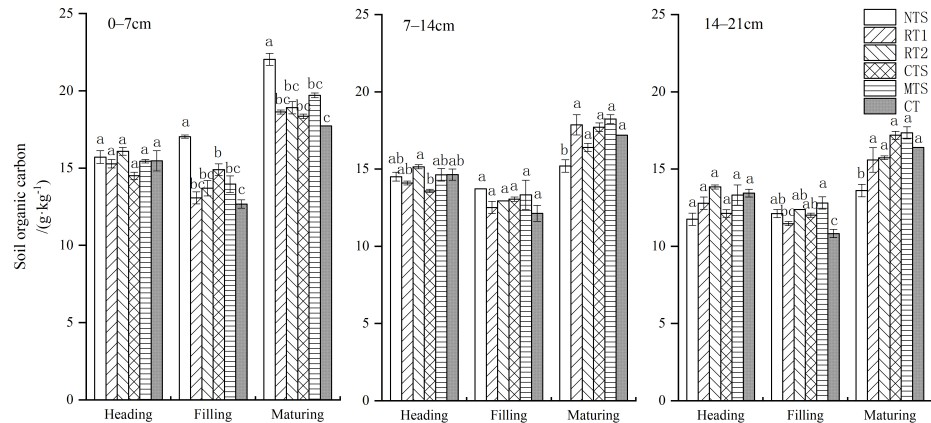

**Figure 4.** The mean and standard error of soil organic carbon (SOC) for different treatments during the late growth stages of rice (heading, grain filling, and maturity). The tillage acronyms are NTS = no-tillage with all straw returning, RT1 = wheat plow tillage and rice no-tillage with half straw returning, RT2 = wheat no-tillage and rice plow tillage with half straw returning, CTS = plow tillage with all straw returning, MTS = less tillage with half straw returning, CT = plow tillage with no straw returning. Lowercase letters indicate significant differences among different tillage methods and straw returning amounts.

### 3.5. The Ratio of Microbial Biomasses at Rice Maturity

The ratio of microbial biomass carbon to nitrogen (SMBC/SMBN) basically ranged from 7.0 to 9.0 (Table 2). In the 0–7 cm soil layer, plowing and low-tillage treatments had lower C/N than the no-tillage treatments. In the 7–14 cm and 14–21 cm soil layers, the CT treatment had the lowest C/N. Generally, straw returning increased soil C/N; the NTS and RT1 treatments increased soil C/N to a certain extent; and the CTS and MTS treatments decreased soil C/N.

**Table 2.** Mean of ratio microbial biomass carbon to nitrogen (SMBC/SMBN), microbial quotient (SMQ), and microbial biomass nitrogen to total nitrogen (SMBN/TN) at rice maturity.

| Soil Depth (cm) | Treatments | SMBC/SMBN | SMQ (%) | SMBN/TN (%) |
|---|---|---|---|---|
| | NTS | 7.87 a | 3.58 ab | 8.38 a |
| | RT1 | 7.97 a | 4.02 a | 8.56 a |
| 0–7 | RT2 | 7.83 a | 3.46 bc | 7.82 a |
| | CTS | 7.11 a | 3.31 bc | 7.92 a |
| | MTS | 7.16 a | 3.16 b | 7.76 a |
| | CT | 7.24 a | 2.95 c | 7.84 a |
| | NTS | 7.43 b | 4.22 a | 11.16 a |
| | RT1 | 7.25 b | 3.18 b | 6.90 b |
| 7–14 | RT2 | 7.97 a | 3.04 b | 7.92 b |
| | CTS | 6.86 c | 2.80 b | 6.84 b |
| | MTS | 7.67 ab | 2.78 b | 6.06 b |
| | CT | 6.67 c | 2.08 c | 6.10 b |
| | NTS | 7.46 a | 2.27 a | 5.71 ab |
| | RT1 | 7.68 a | 2.09 a | 4.98 b |
| 14–21 | RT2 | 7.21 a | 2.42 a | 6.39 a |
| | CTS | 7.15 a | 2.46 a | 5.68 ab |
| | MTS | 8.09 a | 2.35 a | 4.76 b |
| | CT | 6.97 a | 1.54 b | 4.52 b |

Lowercase letters indicate significant differences among different tillage methods ($p < 0.05$).

Similar to the ratio of SMBC/SMBN, SMQ followed a descending order in terms of soil depth. SMQ ranged from 3.0% to 4.0% in the 0–7 cm soil layer, from 2.1% to 4.2% in the 7–14 cm soil layer, and from 1.5% to 2.5% in the 14–21 cm soil layer (Table 2). In the

0–7 cm and 7–14 cm soil layers, SMQ of the NTS and RT1 treatments was higher than that of other treatments. In the 14–21 cm soil layer, SMQ of the RT2, CTS, and MTS treatments was higher than that of other treatments. The CT treatment had the lowest SMQ in the whole soil layer.

Furthermore, soil SMBN/TN had a similar trend to SMQ in terms of soil depth. SMBN/TN ranged from 7.8% to 8.6% in the 0–7 cm soil layer, from 6.1% to 11.2% in the 7–14 cm soil layer, and from 4.5% to 6.4% in the 14–21 cm soil layer (Table 2). The NTS treatment had the highest SMBN/TN in the whole soil layer, while there was no significant difference in SMBN/TN of other treatments (Table 2).

### 3.6. Rice Yield and Yield Components

The NTS treatment had the highest ear number, which increased by 21.97% compared with CT. The CTS treatment had the highest grains per spike, with a value of 182.33. The theoretical yields of the NTS and CTS treatments were higher than that of other treatments. The theoretical yield of CTS was increased by 18.83% compared to that of CT, and that of NTS was increased by 4.94%. The actual yields decreased in the order of CTS > MTS > NTS > CT > RT2 > RT1 (Table 3).

**Table 3.** Rice yield and yield components under different treatments.

| Treatments | Ear Number ($10^4$ ha$^{-1}$) | Grains Per Spike | 1000-Grain Weight (g) | Theoretical Yield (kg ha$^{-1}$) | Actual Yield (kg ha$^{-1}$) |
|---|---|---|---|---|---|
| NTS | 295.93 a | 139.57 b | 25.50 a | 9956.84 b | 9145.67 ab |
| RT1 | 247.49 a | 148.00 ab | 25.07 a | 8658.75 b | 7288.13 b |
| RT2 | 247.01 a | 152.57 ab | 25.61 a | 8759.01 b | 8403.87 ab |
| CTS | 262.18 a | 182.33 a | 25.42 a | 11,273.50 a | 9629.31 a |
| MTS | 238.20 a | 172.47 ab | 25.46 a | 9518.11 b | 9515.10 a |
| CT | 242.62 a | 174.17 ab | 25.16 a | 9487.27 b | 8468.85 ab |

Lowercase letters indicate significant differences among different tillage methods ($p < 0.05$).

## 4. Discussion

In this study (Table 1), we set up three tillage methods with three amounts of straw returning to analyze the effects of tillage and straw returning on the characteristics of TN, SOC, SMBC, and SMBN. We found that no-tillage combined with all straw returning (NTS) effectively increased SOC, TN, SMBC, and SMBN in the 0–7 cm soil, suggesting that NTS increased soil fertility microbial activity.

### 4.1. Effects of Tillage and Straw Returning on SMBC, SMBN, TN, and SOC

In this study (Figure 1), we studied the effects of straw returning on SMBC and SMBN in the soils at different layers and found that significantly increased SMBC and SMBN in the soils, especially in the 0–7 cm soil layer. Straw returning provides the farmlands with organic carbon and microorganisms, which can effectively improve the physical and chemical properties of the soils. Straw returning also increases the total nitrogen content, available phosphorus, available potassium, and trace elements in the soils, resulting in increased SMBC and SMBN. In a previous long-term study, Nie et al. [29] reported that rice straw returning enhanced SMBC and SMBN by 7.8% and 31.4% as compared with no straw returning, which supported our study.

The results showed that under the conditions of straw returning, tillage had significant effects on SMBC and SMBN in the soils of different layers. In the 0–7 cm and 7–14 cm layers of soils, no-tillage enhanced the contents of SMBC and SMBN. However, in the 14–21 cm soil layer, it was plowing and less tillage that significantly increased the contents of SMBC and SMBN. This phenomenon may be due to the interaction of straw returning and no-tillage. Long-term no-tillage combined with continuous straw returning made the straw evenly cover the soil 0–7 cm and regulated moisture and heat conditions in the soils. At the same time, continuous no-tillage stabilized the microorganisms in the

0–7 cm soil layer. That was why no-tillage combined with straw returning was beneficial to increase SMBC and SMBN in the 0–7 cm and 7–14 cm soil depth. A stable soil structure is conducive to the growth of microorganisms. On the contrary, conventional plowing destroys the soil granular structure, which may destabilize the living environment and result in the reduction in soil microorganisms. Plowing and less tillage mixed the straw into the 14–21 cm soil layer and brought a certain amount of water to the 14–21 cm soil layer and balanced oxygen there. These could be the reasons that the contents of SMBC and SMBN of the plowing and fewer tillage treatments (CTS and MTS) in the 14–21 cm soil layer were much higher than other treatments.

In addition, we found that the contents of SMBC and SMBN were the highest at grain filling, but declined rapidly at maturity. This trend coincides with the nutrient requirements of rice plants. From heading to grain filling, rice plants are highly nutrient dependent and require a large amount of nutrients. At maturity, the root system's nutrient dependence on soils weakened [30]. At this stage, paddy fields are usually not irrigated, resulting in lower soil moisture content and fewer nutrients available for microorganisms, which in turn leads to a sharp decline in SMBC and SMBN contents.

In most previous studies, no-tillage and less tillage combined with straw returning enhanced the content of TN in soils [31–33]. In our study (Figure 3), the TN content in the 0–7 cm soil was also observed to be higher in the treatment of no-tillage with straw returning, but the no-tillage treatment had no significant effect on the 7–14 cm and 14–21 cm soil layers. Our finding that straw returning had no significant effects on SOC concentration and no-tillage enhanced SOC concentration in the 0–7 cm soil layer was similar to the findings of João et al. [34] and Blanco-Canqui [35]. As our results showed, more C and N could be translated to rice plants and enhanced the ear numbers, leading to higher production.

*4.2. Effects of Tillage and Straw Returning on SMBC/SMBN, SMQ, and SMBN/TN*

SMBC/SMBN is the ratio of microbial biomass C to N, which is usually considered an indicator of soil N mineralization capacity and has important ecological significance for the balance of C and N in the soils. The value of SMBC/SMBN affects the growth of crops. When SMBC/SMBN < 15, microorganisms multiply fast and there is excess N in the soil. When 15 < SMBC/SMBN < 30, C and N in the soils reach a balance. When SMBC/SMBN > 30, the reproduction of microorganisms is limited, and they will compete with organic matter for N in the soils [36] and crop growth is inhibited. In our study (Table 2), although both straw returning and no-tillage enhanced SMBC/SMBN in the soils, it was maintained between 7–9 on the whole, which ensured an effective supply of N (Table 2). Higher N distribution in soil increased soil fertility and provided a good foundation for rice growth at the heading and filling stages.

No-tillage combined with straw returning increased SMQ in the 0–7 cm and 7–14 cm soil layers, while plowing combined with straw returning increased SMQ in the 14–21 cm soil layer. No-tillage treatment had the highest SMBN/TN in the whole soil layer. The reason might be that straw decomposing brought more microorganisms to the soils, which evenly dispersed in the lower layer of the soils by plowing. Meanwhile, no-tillage combined with straw returning fixed C and N in the 0–7 cm soil layer [37,38]. Our results suggested that no-tillage combined with straw returning maintained the health of soil microbial populations and the stability of soil C and N, which caused a positive effect on crop growth.

## 5. Conclusions

We found that tillage and straw returning had significant impacts on the contents of SMBC, SMBN, TN, and SOC. On the one hand, straw returning significantly increased SMBC and SMBN of the cultivated layer (0–21 cm). Under the condition of straw returning, no-tillage increased SMBC, SMBN, TN, and SOC in the 0–7 cm soil layer, while plowing and less tillage increased SMBC and SMBN in the 14–21 cm soil layer. On the other hand, the soil SMBC/SMBN was maintained within a reasonable range between 7 and 9. No-tillage

enhanced soil SMQ and SMBN/TN to a certain extent, indicating that no-tillage increased SMBC and SMBN and maintained N supply capacity and microbial activity in the soils. In conclusion, no-tillage combined with an appropriate amount of straw returning might be a suitable farming practice to improve the ecological environment of farmlands in the region we studied.

**Author Contributions:** Conceptualization, S.L.; methodology, S.L.; software, W.C.; validation, W.C., W.Y. and Z.W.; formal analysis, W.C. and J.W.; investigation, W.C. and Z.Z.; resources, S.L.; data curation, W.C.; writing—original draft preparation, W.C.; writing—review and editing, W.C.; visualization, W.C.; supervision, S.L.; project administration, S.L. All authors have read and agreed to the published version of the manuscript.

**Funding:** This research was funded by the National Key Research and Development Program of China, grant number 2016YFD0200107, National Science and Technology Planning Project, grant number 2015BAD01B03, Science and Technology Support Program of Jiangsu Province, grant number BE2015337, and Priority Academic Program Development of Jiangsu Higher Education Institution, grant number 201812.

**Data Availability Statement:** Data is contained within the article.

**Acknowledgments:** We thank Zhou for his language help. We also thank Zhuang, Huang, and Jiang for their technical support and writing assistance.

**Conflicts of Interest:** The authors declare no conflict of interest.

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
