# Peer review of "No-Tillage Combined with Appropriate Amount of Straw Returning Increased Soil Biochemical Properties"

_sustainability, doi:10.3390/su14094875_

Round 1

Reviewer 1 Report

It is recommended to strictly use SI units. Table 1. - the amount of straw returning, kg/hm2 - adjust.

Page 3 and others - units cm can be used exceptionally, according to the SI system use m or mm.

Author Response

Response to Reviewer 1 Comments

Point: It is recommended to strictly use SI units. Table 1. - the amount of straw returning, kg/hm2 - adjust. Page 3 and others - units cm can be used exceptionally, according to the SI system use m or mm.

Response: We thank the reviewer for raising this question, and we have changed kg/hm2 to kg hm-2 (Table.1).

Reviewer 2 Report

The topic is OK but the manuscript should be improvement a lot before publication. Sume main opinions and suggestion are shown below and more details can be fond in attached file.

Abstract:

  1. The background in abstract should be rewrite.
  2. There should be noted when the first use of abbreviation。
  3. Please reconsider the use of surface layer, middle layer and lower layer.

Introduction

  1. Citation should be added for some sentences.
  2. What is the innovation of the paper?
  3. The research progress should be expand and the current is far more enough. More progress should pay attention to the influence of straw returning and tillage on foil fertility. And the author should prove why you choose these indicators.

Materials and methods

  1. The author should introduce why set 0-7, 7-14, and 14-21 cm ?
  2. The agriculutal managements for each treatment should be introduced.

Results

  1. I suggest the author add grain yield.
  2. The correlation of the indicators are meaningless.
  3. The author should introduce why use these indicaotrs.

Discussion

  1. The discussion is boring and I suggest the author rewrite this part.
  2. Please don't discuss the indicator one by one, not higher or lower, more discussion should pay attention to the soil fertility or quality. 

Author Response

Response to Reviewer 2 Comments

Point 1: Abstract

1.1The background in abstract should be rewrite.

Response: We thank the reviewer for raising this question and we have rewritten the background. (Line 10-13)

1.2There should be noted when the first use of abbreviation.

Response: We have corrected them. (Line 17-23)

1.3Please reconsider the use of surface layer, middle layer and lower layer.

Response: We removed the use of surface layer, middle layer and lower layer after consideration, and replaced them with 0-7 cm, 7-14 cm and 14-21 cm in the manuscript.

Point 2: Introduction

2.1Citation should be added for some sentences.

Response: We thank the reviewer for raising this question and we have added more citations in introduction.

2.2What is the innovation of the paper?

Response: Most scholars only conduct short-term studies on farmland soil carbon and nitrogen in China with a single measure of tillage methods or straw returning. This study set up 6 tillage methods with different amounts of straw, and comprehensively analyzed the effects of different tillage and straw returning on the content of TN, SOC, SMBC, and SMBN.

2.3The research progress should be expand and the current is far more enough. More progress should pay attention to the influence of straw returning and tillage on foil fertility. And the author should prove why you choose these indicators.

Response: We have added in the introduction. This study was a one-year field experiment at a long-term experiment which did not involve the measurements of soil biochemical properties before 2020. So we set up this study to analyze the effects of tillage and straw returning on the characteristics of TN, SOC, SMBC, and SMBN. (Line 71-74)

Point 3: Materials and methods

3.1The author should introduce why set 0-7, 7-14, and 14-21 cm?

Response: We thank the reviewer for raising this question and we have answered the question in the materials and methods. The cultivated layer was generally 20cm. The depth of less tillage was 8-10 cm, and that of plow tillage was 14-16 cm. Therefore, the cultivated layer was divided into 0-7 cm, 7-14 cm and 14-21 cm for research. (Line 117-119)

3.2The agricultural managements for each treatment should be introduced.

Response: We have added in the materials and methods. The agricultural managements of all the treatments, including fertilizer application, irrigation, weed and pest control, were conducted in conformity with local recommendations. We were advised to apply N at the rate of 245 kg hm-2 from urea, P at the rate of 600 kg hm-2 from calcium superphosphate and K at the rate of 120 kg hm-2 from potassium chloride before sowing. And 147 kg urea hm-2 should be applied in the late growth stage of rice. (Line 109-112)

Point 4: Results

4.1I suggest the author add grain yield.

Response: We thank the reviewer for raising this question and we have added grain yield, as shown in the Table 3.

Table 3. Rice yield and yield components under different treatments.

Treatments

Ear Number

Grains Per Spike

1000-Grain Weight

Theoretical Yield

Actual Yield

(104 hm2)

(g)

(kg hm-2)

(kg hm-2)

NTS

295.93a

139.57b

25.50a

9956.84b

9145.67ab

RT1

247.49a

148.00ab

25.07a

8658.75b

7288.13b

RT2

247.01a

152.57ab

25.61a

8759.01b

8403.87ab

CTS

262.18a

182.33a

25.42a

11273.50a

9629.31a

MTS

238.20a

172.47ab

25.46a

9518.11b

9515.10a

CT

242.62a

174.17ab

25.16a

9487.27b

8468.85ab

4.2The correlation of the indicators are meaningless.

Response: We have removed it.

4.3The author should introduce why use these indicaotrs.

Response: We have explained it in the introduction. (Line 71-74)

Point 5: Discussion

5.1The discussion is boring and I suggest the author rewrite this part.

Response: We have revised the discussion.

5.2Please don't discuss the indicator one by one, not higher or lower, more discussion should pay attention to the soil fertility or quality. 

Response: We have revised the discussion.

Reviewer 3 Report

The manuscript entitled “No-Tillage Combined with Appropriate Amount of Straw Returning Increased Soil Biochemical Properties” (reference sustainability-1663558) studied the effects of tillage and straw returning operations on soil biochemical properties through field experiment. The topic is meaningful for agricultural production and straw management. However, some points need to be further discussed or clarified before it is suitable for publication. Comments are shown as below:

Line 10: As I understand, this is a one-year experimental study. How can you say to investigate the long-term effects of tillage and straw returning?

Lines 16-20: Please indicate the meaning of SMBC, SMBN, SMQ, SOC etc. in the abstract.  

Lines 61-62: You mentioned previous studies were mostly short-term studies, so do you mean this study can be viewed as a long-term study? How?

Lines 114-115: Please check, I think the SMQ should be “the ratio of SMBC to SOC”?

Lines 304-312: Are you discussing the SMBC/SMBN or C/N, or both in this paragraph. The current description makes me confused.

Line 316: Reference [24] should be specified here.

Results and Discussions part: Straw returning amount was also a variable in the experimental design. However, very few discussions are about the returning amount and a very vague conclusion about this was made as “appropriate amount of straw returning” (line 333) in the Conclusion part. More detail analysis and discussions on the straw retuning amount should be added.

Author Response

Response to Reviewer 3 Comments

Point 1: Line 10: As I understand, this is a one-year experimental study. How can you say to investigate the long-term effects of tillage and straw returning?

Response: We thank the reviewer for raising this question and we have corrected it.

Point 2: Lines 16-20: Please indicate the meaning of SMBC, SMBN, SMQ, SOC etc. in the abstract.  

Response: We thank the reviewer for raising this question. We have corrected the use of abbreviation. (Lines 17-23)

Point 3: Lines 61-62: You mentioned previous studies were mostly short-term studies, so do you mean this study can be viewed as a long-term study? How?

Response: We thank the reviewer for raising this question. This study was conducted during the rice growing season of 2020 (from May 2020 to October 2020) at a long-term experimental location starting in November 2001 in the experimental field of Yangzhou University. We measured the NPK nutrient content and other soil fertility every season to compare it with the basal soil fertility in 2001. But this study did not mention the measurements.

Point 4: Lines 114-115: Please check, I think the SMQ should be “the ratio of SMBC to SOC”?

Response: We thank the reviewer for raising this question. We have corrected it. (Line 127)

Point 5: Lines 304-312: Are you discussing the SMBC/SMBN or C/N, or both in this paragraph. The current description makes me confused.

Response: We thank the reviewer for raising this question. We discussed the SMBC/SMBN in lines 338-340 and we have corrected it.

Point 6: Line 316: Reference [24] should be specified here.

Response: We thank the reviewer for raising this question and we have added. (Line 347 and 348)

Point 7: Results and Discussions part: Straw returning amount was also a variable in the experimental design. However, very few discussions are about the returning amount and a very vague conclusion about this was made as “appropriate amount of straw returning” (line 333) in the Conclusion part. More detail analysis and discussions on the straw retuning amount should be added.

Response: We thank the reviewer for raising this question. The amount of straw returning was total amount of rice straw (4500kg hm-2) and less amount of rice straw (3000 kg hm-2). But in this study, straw returning only increased SMBC and SMBN, and had no significant effects on TN and SOC. The response of soil biochemical properties to tillage was more significant. So we could only draw the conclusion that straw returning was beneficial to improving the soil quality.

Round 2

Reviewer 2 Report

The manuscript has improved a lot compared to the original one. However, I have to say, every published article should be taken seriously, the paper still have a lot of small mistakes, some should be modified the following ones before publicaiton and more should be checked by the authors carefully:

  1. Line 89-94 should be moved to measurements and, actually, they are repeated with Line 120-122.
  2. The problem of superscripts and subscripts still exists, such as Line 124, please check it carefully from beginning to end. 
  3. Please use unified unit for the manuscript such as in Line 200, the prrevious one use g/kg and the last one use mg/g for SOC.
  4. Please consider change hm-2 to ha for the manuscript. 
  5. The discussion is still weak.

Author Response

Response to Reviewer 2 Comments

The manuscript has improved a lot compared to the original one. However, I have to say, every published article should be taken seriously, the paper still have a lot of small mistakes, some should be modified the following ones before publicaiton and more should be checked by the authors carefully.

Point.1 Line 89-94 should be moved to measurements and, actually, they are repeated with Line 120-122.

Response: We thank the reviewer for raising this question and we have removed Line 89-94.

Point.2 The problem of superscripts and subscripts still exists, such as Line 124, please check it carefully from beginning to end. 

Response: We thank the reviewer for raising this question. We have corrected it and checked the full manuscript.

Point.3 Please use unified unit for the manuscript such as in Line 200, the prrevious one use g/kg and the last one use mg/g for SOC.

Response: We thank the reviewer for raising this question. The unit of SOC is g/kg and we have corrected the wrong unit.

Point.4 Please consider change hm-2 to ha for the manuscript. 

Response: We thank the reviewer for raising this question and we have changed hm-2 to ha.

Point.5 The discussion is still weak.

Response: We thank the reviewer for raising this question. But different treatments had different effects on each indicator, we hesitate to discuss all indicators together. This seems a little confusing. So we have simplified the discussion of each indicator and added the discussion of soil fertility to make it clearer.